# Asthma Care from Home: Study protocol for an effectiveness-implementation evaluation of a virtually enabled asthma care initiative in children in rural NSW

Ryan Mackle[1,2☯], Carmen Crespo Gonzalez[1☯], Mei Chan[1], Michael Hodgins[1], Nan Hu[1], Blake Angell[3], Louisa Owens[1,2], Jeffery Fletcher[4,5], Timothy McCrossin[6,7], Susie Piper[8], Aunty Kerrie Doyle[9], Sue Woolfenden[1,10,11], Bronwyn Gould[12], Flic Ward[13], Raghu Lingam[1,14], Adam Jaffe[1,2], Melinda Gray[2], Nusrat Homaira[1,2]*, on behalf of the Asthma Care from Home Collaborative Group[¶]

1 Discipline of Paediatrics and Child Health, Faculty of Medicine, UNSW Sydney, Sydney, NSW, Australia, 2 Respiratory Department, Sydney Children's Hospital Randwick, Sydney Children's Hospital Network, Sydney, NSW, Australia, 3 The George Institute for Global Health, UNSW Sydney, NSW, Sydney, Australia, 4 Department of Paediatrics, The Tweed Hospital, Northern NSW Local Health District, Tweed Heads, Australia, 5 School of Medicine, Griffth University, Gold Coast, QLD, Australia, 6 Department of Paediatrics, Bathurst Base Hospital, Western NSW Local Health District, Bathurst, NSW, Australia, 7 Bathurst Rural Clinical School, School of Medicine, Western Sydney University, Sydney, NSW, Australia, 8 Department of Paediatrics, South East Regional Hospital, Southern NSW Local Health District, Bega, NSW, Australia, 9 Discipline of Indigenous Health, School of Medicine, Western Sydney University, Sydney, NSW, Australia, 10 Faculty of Medicine and Health, University of Sydney, Sydney, NSW, Australia, 11 Department of Community Paediatrics, Sydney Local Health District, Sydney, NSW, Australia, 12 General Practice, Paddington, NSW, Australia, 13 Parent of Child with Asthma, Australia, 14 Department of Community Paediatrics, Sydney Children's Hospital Randwick, Sydney Children's Hospital Network, Sydney, NSW, Australia

☯ These authors contributed equally to this work.
¶ Membership of the Asthma Care from Home Collaborative Group is provided in the Acknowledgments.
* n.homaira@unsw.edu.au

**Data Availability Statement:** No datasets were generated or analysed during the current study. All

## Abstract

### Background

Asthma is the leading source of unscheduled hospitalisation in Australian children, with a high burden placed upon children, their parents/families, and the healthcare system. In Australia, there are widening disparities in paediatric asthma care including inequitable access to comprehensive ongoing and planned asthma care for children.

### Methods

The Asthma Care from Home Project is a comprehensive virtually enabled asthma model of care that aims to a. supports families, communities and healthcare providers, b. flexible and locally acceptable, and c. allow for adoption of innovations such as digital technologies so that asthma care can be provided "from home", reduce potentially preventable asthma hospitalisation, and ensure satisfaction at a patient, family, and healthcare provider level. The model of care includes standardisation of discharge care through provision of an asthma discharge resource pack containing individual asthma action plan, follow-up letters for the

relevant data from this study will be made available upon study completion.

**Funding:** This project was funded by a grant from the NSW Health Translational Research Grants Scheme. The views expressed are those of the authors and not necessarily those of the partner organisation. RM is supported by PhD scholarships from the Australian government Research Training Program (RTP), Asthma Australia and the Zoe Kennedy Foundation. The funders had no role in study design, data collection and analysis, decision to publish, or preparation of the manuscript.

**Competing interests:** The authors have declared that no competing interests exist.

**Abbreviations:** AAIC, Aiming for Asthma Improvement in Children; AAP, Asthma Action Plan; ACT, Asthma Control Test; C-ACT, Childhood Asthma Control Test; CFIR, Consolidated Framework for Implementation Research; BMQ, Brief Medication Questionnaire; ED, Emergency Department; eMR, Electronic Medical Record; GEE, Generalized Estimating Equations; GP, General Practitioner; LHD, Local Health District; miniPAQLQ, Mini Paediatric Asthma Quality of Life Questionnaire; NSW, New South Wales; SCH, Sydney Children's Hospital; SMS, Short Message Service.

child's general practitioner (GP) and school/child care, and access to online asthma educational sessions and resource; post-discharge care coordination through text message reminders for families for regular GP review, email correspondence with their child's GP and school/childcare; and virtual home visits to discuss home environmental triggers, provide personalised asthma education and respond to parental concerns relating to their child's asthma. This study is comprised of three components: 1) a quasi-experimental pre/post impact evaluation assessing the impact of the model on healthcare utilisation and asthma control measures; 2) a mixed-methods implementation evaluation to understand how and why our intervention was effective or ineffective in producing systems change; 3) an economic evaluation to assess the cost-effectiveness of the proposed model of care from a family and health services perspective.

## Discussion

This study aims to improve access to asthma care for children in rural and remote areas. Implementation evaluation and economic evaluation will provide insights into the sustainability and scalability of the asthma model of care.

## Background

Asthma is a complex disease and remains one of the most common disorders worldwide [1]. Our improved understanding of asthma has led to advances in management and asthma outcomes have improved, particularly in high income countries like Australia. However, health disparities endure as children continue to have a high associated burden of care, high levels of uncontrolled asthma, and increased healthcare utilisations [2]. Within Australia, 1 in 10 children have a diagnosis of asthma, and it leads to more than 20,000 unscheduled hospital presentations in children with asthma, costing the health system more than $300 million annually [3]. This high rate of hospitalisation is not only associated with high costs to the healthcare system but also results in school absenteeism, significant burden on families from days missed from work and psychological stress [4].

Children living in rural and remote settings are particularly disadvantaged and face multiple barriers to asthma care. The highest burden of care and mortality rates from asthma in Australia are reported in rural and remote communities despite having lower prevalence rates compared to their urban counterparts [5]. In the state of New South Wales (NSW), there are around 50,000 children with asthma living in rural and remote regions [6, 7]. Studies from the US suggest rural children are significantly more likely to have moderate to severe asthma compared to urban children (46% vs 35%) [8]. Similar studies are yet to be done in Australia on a large scale. In addition, rural families have on average 20% less disposable income than families from urban areas, they have less access to primary care, journeys to seek healthcare are eight times longer, and more time is spent absent from work or school due to healthcare appointments compared to urban-based families [7, 9].

At the core of asthma management is the focus on establishing effective asthma control and risk reduction of morbidity and mortality. Traditionally, asthma management in children has required regular attendance to healthcare professionals (General Practitioners (GPs) Paediatricians and/or Respiratory specialists) for review of asthma control, monitor lung function parameters, asthma education, assess medication adherence and response, and titration of

pharmaceutical therapy as necessary [10]. The recent COVID-19 pandemic has however enforced changes in clinical practice both in Australia and internationally at short notice, with less face-to-face interactions, increased use of emergency departments (EDs) for primary care, and a new reliance on technology to ensure continued access to care. With the emergence of e-health and digitally based interventions, the COVID-19 pandemic has highlighted and quickened the use of these interventions in daily practice, however its specific role in paediatric asthma care needs more thorough evaluation [10].

The current model of care for childhood asthma in Australia, which is hospital-centred, is fragmented, complex and inadequate [11]. Our prior work has demonstrated several gaps in childhood asthma care in Australia. Firstly, Australian children receive guideline adherent asthma care on around 60% of occasions [12]. Within NSW, post-discharge asthma management pathways vary between local health districts (LHDs), between hospitals in the same LHD and within departments of the same hospital. On average four to six different asthma clinical practice guidelines and asthma action plans were used in each LHD [13]. Additionally, asthma education provided in hospitals was often non-formal and rarely involved key topics such as knowledge of asthma, asthma control and regular medical review. Lastly, there is currently no system to ensure subsequent follow-up with General Practitioners (GPs) after discharge from hospitals and linkages with community-based services are almost non-existent [13], highlighting the need for a comprehensive integrated model of paediatric asthma care.

Systematic review and meta-analysis from our team have shown that a multicomponent comprehensive asthma model of care that includes i) asthma self-management education, ii) active care coordination connecting patients/families with acute, primary and community-based services, and iii) in-home environmental assessment can reduce asthma hospital presentations by 80% [11]. Subsequently we evaluated a co-developed integrated model of asthma care, co-designed with parents and professionals, at Sydney Children's Hospital (SCH), which included the provision of a standardised asthma discharge pack for children and parents containing an individualized Asthma Action Plan, information pack and discharge instructions (including recommended follow-up with a GP), streamlined appointments with a GP post-discharge from the hospital, and text message reminders to parents reminding them to follow-up with GP and encouraging attendance to an asthma education session. This model of care led to a 56% reduction in number of children who presented frequently to the ED with asthma [14]. Although the model of care was effective in a large metropolitan children's hospital, its value in rural areas cannot be assumed and translation of the successful model into rural areas requires robust evaluation.

Therefore, leveraging digital technology and building on our existing work on paediatric asthma, we will implement and evaluate a comprehensive and flexible model of care that ensures standardisation of asthma discharge from hospital, active care coordination between acute and primary care services, establishes linkage with a child's school/childcare services and delivers home-based follow up services to provide optimal health outcomes in children with asthma living in rural NSW.

## Hypotheses and aims

We hypothesise that a comprehensive virtually enabled asthma model of care that leverages digital technologies can provide continuity of care, improve asthma quality of life and lead to reduction in unscheduled hospital presentations in children with asthma aged 5–12 years living in rural NSW. We further hypothesise that our proposed model of care will reduce the cost of care devoted to this patient cohort and help to alleviate financial pressures facing families as a result of a child's asthma diagnosis.

Asthma Care from Home aims to evaluate the impact (aim 1), implementation (aim 2) and cost effectiveness (aim 3) of a comprehensive virtually enabled asthma care initiative in three rurally based LHDs in NSW that a. supports families, communities and healthcare providers, b. is flexible and locally acceptable, and c. adopts innovations such as digital technologies so that asthma care can be provided 'from home' when necessary, thereby not only reducing the significant burden of potentially preventable asthma hospital presentations but also ensuring patients', parents/carers and families satisfaction with healthcare delivery.

This project comprises of three related research questions:

1. What is the impact on asthma-related health system utilisation, child's health related quality of life and parent/carer's experiences?

2. What is the cost effectiveness and cost impact for both the child's family and the healthcare system by adopting this model?

3. Can these results be used to support State-wide scale-up?

## Study methods

### Study design and setting

Our study will be an integrated impact, implementation and economic evaluation embedded within a quasi-experimental pre and post interventional trial at three large rural Local Health Districts (LHDs) in NSW, Australia: Northern NSW Local Health District (NNSWLHD), Southern NSW Local Health District (SNSWLHD) and Western NSW Local Health District (WNSWLHD). Ethical approval was granted by the Sydney Children's Hospital Network Human Research Ethics Committee in November 2022 (2022/ETH01946). Site specific Research Governance was granted across all participating Local Health District Research Governance by March 2023 (2022/STE03422, 2022/STE03423, 2022/STE03424 and 2022/STE03425).

### Study population

Between March 2023 and June 2024, children aged between 5–12 years old presenting to ED on one or more occasions or admitted to the general paediatric ward on one or more occasion in the last 12 months with asthma at selected study hospital sites within the three LHDs will be eligible to participate in the study. For those that fulfil all eligibility criteria (Box 1), each

---

### Box 1

Inclusion criteria

- Children aged between 5–12 years old.

- 1 or more ED presentations or 1 or more hospital admission in the last 12 months

- Children with current or history of physician-diagnosed asthma

- Children attended EDs or admitted to general paediatric wards with asthma exacerbation or asthma-related illnesses (e.g. wheeze) at hospitals within the three participating LHDs during the study period

---

- Parents/carers of non-English speaking background but understand English and do not require interpreter.

Exclusion criteria

- Children with complex conditions e.g., genetic disorders, complex cardiac diseases etc.

- Children with high-risk and complex asthma who are already known to tertiary paediatric respiratory specialists.

- Children with significant developmental delays or comorbidities that will significantly limit their ability to participate in the study.

- Both parents/carers who do not understand English and require interpreter

- Families who do not have permanent housing e.g., live in a shelter.

participant will be followed for a period of 12 months after enrolment. Informed written consent was sought from parents/carers of eligible children electronically via REDcap software.

## Study interventions

Eligible participants (Box 1) will receive a suite of interventions adapted from our previous work [14]. This will be in addition to current expected "standard" discharge care including a salbutamol weaning plan, medication prescriptions (when necessary), and hospital discharge letters as recommended in NSW Health guidelines [15]. The interventions include:

**1. Standardisation of asthma discharge.** Participants discharged from the hospital will receive the "Asthma Resource Discharge package" which contains several pre-developed resources including an individualised asthma action plan (AAP) in colour [16], an asthma education booklet entitled "Asthma and Your Child–a resource pack for parents and carers" to improve parent's/carer's understanding of asthma that is available in multiple languages, a parent letter encouraging regular GP follow-up and uptake of annual influenza vaccination (as recommended in Global Initiative for Asthma (GINA) guidelines [17] and the Australian Immunization Handbook [18]), a letter for the child's GP advising of their recent hospital presentation and discharge instructions including clinical best practice asthma tips, a letter for the child's school/childcare including an individualised "Schools and Child Services Action Plan for Asthma Flare-up" [19] as approved by NSW Health and NSW Department of Education, an asthma leaflet with QR code link to access online based multimedia educational resources and access to monthly educational webinars facilitated by asthma clinical nurse consultants in the Aiming for Asthma Improvement in Children Initiative (AAIC, an NSW Health statewide asthma education organisation, www.asthmainchildren.org.au).

**2. Post-discharge care coordination.** Parents/carers of the participating child will firstly receive a text message (SMS) reminder within 7 days of hospital discharge to remind them of regular GP follow-up and provide links to online resources and links to register for monthly educational webinars with the AAIC. In addition, local investigators will coordinate follow-up appointments with GPs and/or paediatricians through telephone or email to facilitate tele-consultation where possible and provide discharge instructions. Similarly, the child's school or childcare service will be notified by email of the child's recent asthma hospitalisation, provided an updated School & Child Services Action Plan [19] and reminded of the availability of online staff training resources including the AAIC eBook on Asthma First Aid in Children or training

sessions for childcare staff via ACECQA (Australian Children's Education & Care Quality Authority).

**3. Follow-up support for children with uncontrolled asthma.** In addition to SMS messaging, children with uncontrolled asthma, as evidenced by an ACT/C-ACT score <19 on baseline questionnaire at enrolment, will be offered virtual home visits within 3 months of discharge from hospital to work through home environmental triggers, provide personalised asthma education and respond to parent's concerns relating to their child's asthma management plan. Assessment of environmental triggers including aeroallergens like house dust mite, pollens and grasses is important as they are key contributors to asthma disease and can result in frequent flare-ups [20]. Virtual home visits will be based on the "home environment checklist" (adapted from the home visit checklists of the Royal Brompton & Harefield Hospital and the United States Environmental Protection Agency) and performed by local clinical staff. These virtual home visits are a novel intervention within the three study LHDs. They will occur remotely using encrypted digital platforms and last for approximately 20–30 minutes.

For children identified as having severe asthma, defined by GINA as "asthma that is uncontrolled despite adherence with maximal optimized therapy and treatment of contributory factors, or that worsens when high dose treatment is decreased" [17], will also be offered joint telehealth consultation with a tertiary hospital-based paediatric respiratory consultant at the Sydney Children's Hospital, Randwick and a locally based general paediatrician to develop a shared care plan. This is in keeping with National Asthma Council guidelines, which recommend that those on step 4 of asthma management algorithm be considered for specialist review [21].

## Aim 1: Impact assessment

**Data collection.** Changes in health service use for participating children will be evaluated by comparing the period of up to 5 years prior to their enrolment into the program (pre-intervention) and the period of up to 12 months following enrolment (post-intervention). This information will be gathered primarily from electronic care records at baseline (retrospectively gathering data on health service use in previous 5 years) and 12 months post-enrolment. Five years data pre-enrolment will be used to account for disruption from the COVID-19 pandemic.

As we are following up participants through time in each site, the pre- intervention time period of health service use for each participating site will serve as its own control for the post-intervention time period of health service use, that is, temporal control.

*Outcomes measures.* All outcome measures for the impact assessment will be assessed at baseline, 6 months, and 12 months post-intervention (Table 1). We will measure the changes over time (i.e., pre-intervention at baseline and post-intervention at 6-month and 12-month), and differences between intervention and temporal control groups on all indicators.

*Demographics.* Key demographic data including age, gender, language spoken at home, country of birth, parent/carer level of education, Socio-Economic Indexes for Areas (SEIFA) data and indigenous status will be collected from electronic medical records and the baseline questionnaire.

*Healthcare utilisation.* Researchers will focus on 3 key areas of healthcare utilisation: i) GP visits, ii) ED visits, and iii) hospital admission. This will be collected from electronic records and questionnaires.

*Asthma control.* Level of asthma control will be assessed using the Asthma Control Test (ACT; for children $\geq$ 12 years old) or Childhood Asthma Control Test (C-ACT; children 5–11 years old). These validated tools consist of 5–7 Likert-type scale questions, with a 4-week recall, assessing the frequency of shortness of breath and general asthma symptoms, use of rescue

**Table 1. Outcome measures of the impact assessment.**

| Outcome | Measure(s) | Source | Time point |
|---|---|---|---|
| **Primary outcome** | | | |
| Health service use | • Hospital admissions<br>• ED attendances<br>• GP visits | Linked routinely collected medical records | Pre and post intervention |
| **Secondary outcome** | | | |
| Asthma related quality of life in children | miniPAQLQ | Online questionnaire | Enrolment, 6 and 12 months |
| Asthma control | ACT or C-ACT | Online questionnaire | Enrolment, 6 and 12 months |
| Medication adherence | BMQ | Online questionnaire | Enrolment, 6 and 12 months |
| Patient and family reported experiences<br>*Parental satisfaction*<br>*travel, costs, missed school/work, asthma-self management knowledge* | Purpose designed questionnaire | Online questionnaire | Enrolment, 6 and 12 months |

medication, the effect of asthma on daily functioning, and overall self-assessment of asthma symptom control. The total score range is 0–27 (C-ACT) and 0–25 (ACT) respectively, with a score on either test less than 19 indicating uncontrolled asthma [22].

*Quality of life*. Quality of life of the child will be assessed by using the miniPAQLQ validated tool [23], which consists of 13 questions in three domains (activity limitation, symptoms, and emotional function). The activity limitation domain includes 3 questions relating to play, sports and other daily activities; symptoms domain has 6 questions on cough, wheezing and nocturnal awakening; emotional function domain contains 4 questions associated with items such as being frightened, frustrated or feeling different, being irritable or worried, etc. Scores for each item range from 1 (maximum impairment) to 7 (no impairment). Overall score will be calculated by dividing the total scores with 13.

*Medication adherence*. Medication adherence will be assessed by using the Brief Medication Questionnaire (BMQ) [24] which consists of 12 screening questions of three domains (medication regimen, belief, and recall) with 1-week recall. The regimen domain has five items, with scores from 0 (adherence) to 1 (potential non-adherence). The belief domain has 2 items, with scores from 0 (no belief barrier) to 2 (belief barriers present). The recall domain has five items, with scores from 0 (no recall barriers) to >1 (recall barriers present). The questionnaire assesses patient adherence and barriers to adherence.

*Oral corticosteroid use*. Parents/Carers will be asked to recall retrospectively the number of days their child with asthma has required oral corticosteroids (e.g., prednisolone) in the preceding 6 months. Where possible, corroboration will be obtained on review of their electronic medical records if oral corticosteroids have been prescribed within a hospital setting.

**School/Work absence.** Parents/Carers will be asked at 6-month intervals to recall number of days their child has been absent from school/childcare or they have been absent from work due to their child's asthma

*Sample size calculation*. Based on existing literature that shows 2.8% of paediatric ED attendances is asthma-related, we will be able to recruit at least 317 children from three LHDs for the impact assessment. Allowing for 15% attrition this will assure 270 participants for final analysis to detect a rate ratio of 2.6 in the mean number of asthma-related ED presentations before and after intervention, 80% power and 5% significance (two-sided). This calculation was based on the power calculation method for rate ratio of Poisson-distributed count data for a single sample setting.

Participants will be recruited from all eligible sites within each LHD. Recruitment of participants from each LHD will be proportional to the eligible catchment population within the specific LHD.

**Data analysis.**   Interrupted time series (ITS) analysis will be used to assess the change in the time series patterns (intercept and slope) of health service use before and after intervention.

ACT score, asthma related quality of life, medication adherence, parent and child quality of life and family reported experiences will be compared at enrolment, 6 and 12 months after enrolment into the service. We will describe the outcomes at each time point using mean, standard deviation, median, and inter-quartile range. Since each participant will have three repeated measures over time, we will use multi-level regression analysis to examine the effect of the intervention on these outcomes, including in the models, a fixed effect for the intervention status (0 = pre-intervention, 1 = 6-month post-intervention, 2 = 12-month post-intervention) and calendar time of enrolment into the intervention for each participant, as well as a random effect of the Local Health District (LHD) to explore the degree of variation in the intervention effect by LHD. Our final model will exclude the random term in the presence of a statistically non-significant random effect. We will also consider participants are clustered within each LHD and use generalised estimating equations (GEE) approach in the regression analysis to account for the clustering.

## Aim 2: Implementation evaluation

In addition to the Asthma Care from Home impact assessment, we will undertake a mixed-methods implementation evaluation using the evidence-based Consolidated Framework for Implementation Research to understand how and why our intervention was effective or ineffective in producing systems change [25]. We also aim to identify contextually relevant strategies for successful implementation as well as practical difficulties in the adoption, delivery, and maintenance of Asthma Care from Home to inform wider state and potentially national roll-out. The evaluation will allow us to assess implementation metrics as defined by Proctor et al (Table 2) [26].

**Logic model.**   We have created a logic model to help develop and guide the implementation evaluation mixed methods approach to data collection (S2 Fig). The model encompasses the specific contextual factors the implementation evaluation would need to consider both within (e.g., inner context representing individual factors and organisational settings) and external to the sites (e.g., area demographics, socio-economic status) [27, 28].

In addition to the contextual factors, we have attempted to represent the measurable intervention characteristics, including the number of scheduled follow-up virtual consultations, the

**Table 2. Implementation metrics.**

|  | Questions addressed by each implementation factor. |
|---|---|
| **Acceptability** | Do practitioners, parents and children view Asthma Care from Home as agreeable? |
| **Adoption** | To what extent do practitioners and parents use resources? |
| **Appropriateness** | Do stakeholders perceive Asthma Care from Home as relevant & useful? |
| **Fidelity** | Are all component parts of the intervention delivered as planned? |
| **Feasibility** | Are Asthma Care from Home component parts practical to deliver within the service? |
| **Coverage** | How many service users of those eligible are reached? |
| **Cost** | How much does it cost to implement Asthma Care from Home successfully? |
| **Sustainability** | What factors will allow Asthma Care from Home to be scaled-up further? |

extent of administrative support including regular meetings with the project team and the use of project resources. The practical elements of the intervention are underpinned by theoretical principles including the 7 Critical success factors for a successful transition from hospital to home framework and the behaviour change wheel [29, 30]. Finally, we have drawn connections from these underlying theories of change to the specific outcomes we hypothesise the intervention will produce.

*Outcome measures.* Table 3 provides a summary of the outcome measures for the implementation evaluation.

**Data collection.** Data collection will consist of: 1) Routinely collected data on enrolments, diagnoses, and demographic information; 2) Surveys with parents/carers and practitioners involved in clinical completed regarding their experiences of the service; 3) Qualitative data through focus groups and interviews with parents/carers, healthcare providers and project team members.

*Surveys.* Surveys will be collected online via REDCap, a secure web application for building and managing online surveys hosted on UNSW infrastructure.

*Families/carers.* Families/ carers at the end of the study will be asked to complete an evaluation survey to collect their perspectives on the program (Table 3). The survey will include a

**Table 3. Outcome measures of the implementation evaluation.**

| Outcome | Methods and Measures | Participants | Time |
|---|---|---|---|
| Description of local context and practice | Socio-Economic Indexes for Areas (SEIFA) data, search of grey literature, informal contact with LHDs, project data collection logs | Project team | Ongoing throughout implementation |
| Fidelity to the model | Case log.<br>To determine how the intervention is being carried out in each site as compared to the intended rollout, we will draw on trial data, which will record the specific details of model delivery. This will include recording:<br>•Number of eligible patients participating in the study<br>•Number of SMS send to families after discharge.<br>•Number/frequency of virtual home visits conducted.<br>•Attendance to webinars. | LHDs representative and project team members. | Ongoing throughout implementation |
| Barriers and facilitators to running the Asthma Care from Home Model | To determine an individual's knowledge and beliefs about the model of care; relative advantages of the model of care; barriers and facilitators affecting the delivery of the intervention both from an individual and organisational perspective; the appropriateness and acceptability of the intervention; and recommendations for future implementation, we will conduct semi-structured interviews and focus groups with all participant groups at various stages during the trial. Qualitative interviews and focus groups, are guided by the Consolidated Framework for Implementation Research (CFIR) | Practitioners involved in clinical care delivery, families/carers, managers and project team. | Ongoing throughout implementation |
| The Acceptability, Appropriateness, and Feasibility of the model | Acceptability of intervention measure [31], intervention appropriateness measure (IAM), and feasibility of intervention measure (FIM). To measure of the perceived fit, relevance, or compatibility of evidence-based practice for a context, person, or problem. | Practitioners involved in clinical care delivery. | Implementation end (included in the 12-month practitioners involved in clinical care survey) |
| Practitioners buy into the model | To assess to what extent practitioners "buy into" the Asthma Care from Home model of care, and how it becomes part of routine practice, we will use the NoMAD Tool based on normalisation process theory. The NoMAD tool will assess how the intervention was incorporated into standard work responsibilities | Practitioners involved in clinical care delivery. | Implementation end (included in the 12-month practitioners involved in clinical care survey) |
| Family experience and acceptability of the model | The theoretical framework of acceptability (TFA) to evaluate patient acceptability of the model. | Families and carers | Implementation end (included in the 12-month family survey) |

validated tool, based on the theoretical framework of acceptability (TFA) of health interventions tool [32]. Items will be rated on a 5-point Likert scale.

*Practitioners involved in clinical care.* All practitioners involved in clinical care delivery will be asked to complete a short online survey about their experiences with the program. The survey will include two validated instruments: the short intervention acceptability, appropriateness and feasibility measure, and the NOMAD tool (Table 3) [33–35]. Items will be rated on a 5-point Likert scale from 1 (strongly agree) to 5 (strongly disagree).

*Interviews and focus groups.* Interviews with families and carers will determine their perceptions of the acceptability of the model and any potential adaptations to the model to make it more acceptable for children and their families. Practitioners involved in clinical care and managers will specifically be asked about the features of follow up support and coordination with other professionals. Interviews with project team members will provide details about the model implementation process from an implementer's perspective.

**Sample size.**   Participants of our implementation evaluation include consenting practitioners involved in clinical care, families of 5–12-year-olds with asthma, managers and project team members.

**Surveys.**   *Families/carers.* The family experience surveys will be offered to all families actively participating in the Asthma Care from Home program. Information regarding the survey will be sent to all participants at the end of the impact assessment period at 12 months to ensure the anonymity of the families. We expect to collect approximately 100 surveys across the three LHD during the whole intervention period (24 months). No power calculation has been done as we will recruit as many families as possible during this timeframe and utilise all available data.

*Practitioners involved in clinical care.* All practitioners involved in clinical care delivery will be recruited to complete a short online survey after the intervention rollout. A research team member will contact them via email with the information regarding the survey with a direct link to the survey.

*Interviews and focus groups.* Purposive sampling will be used to recruit a diverse sample of service users and providers for the interview and focus groups. Data collection for the qualitative interviews and focus groups will continue until data saturation- no new themes pertaining to the research objectives are identified with subsequent interviews. However, from previous experience, we will contact at least 40 practitioners, managers and families/carers across the three sites, ensuring diversity with regard to practice and role [36].

Families/carers and practitioners who have participated in the model of care will be recruited during the survey intervention period, including an item seeking permission to contact them about the opportunity to participate in a qualitative interview or focus group. The interview can be conducted either in person, online via video or via telephone. We aim to conduct around 20 with practitioners and 20 parents/caregivers across the three LHDs.

Additionally, a research team member will contact managers directly via phone or email to invite them to participate in an interview or focus group and provide information about the objective of the discussion. We aim to recruit at least one manager per LHD.

**Data analysis.**   *Surveys.* In this study, reliability, validity and confidence will be maximised through cross verification and exploration of differences between the qualitative and quantitative, exploring and accounting for differences and mapping the perspectives of different stakeholders across the study. Quantitative questionnaire data will be exported into SPSS/STATA for analysis. Descriptive statistics will be calculated for each LHD in which recruitment is performed including information about inner and outer context, the intervention use and its acceptability. Any open-ended questions will be analysed and, where possible, a coding scheme

will be developed to enable descriptive analysis, and where this is not possible, open-ended questions will undergo inductive thematic analysis.

*Interviews and focus groups*. The study team will audio-record and transcribe interviews and focus groups verbatim and thematically analyse the transcripts to identify, interpret and report on the repeated patterns of meaning within the data, influenced by the CFIR constructs and drawing from Braun and Clark's thematic analysis approach [37]. The data collected will be de-identified in preparation for data analysis and no results will contain any information that could identify participants [37]. Where appropriate, NVivo software will aid in the coding and organisation of themes. An inductive analysis of qualitative data will be conducted to ensure openness to emerging themes not readily captured by the CFIR and Proctor and colleagues' outcome measures. The analysis will specifically focus on key behavioural issues and motivational domains as well as the challenges surrounding intervention implementation.

## Aim 3: Economic evaluation

**Outcome measures.** An economic analysis will determine the relative cost-effectiveness of the implemented model relative to current standard care and will include a cost impact analysis for the LHDs to inform potential future scale-up of the intervention. Results of the cost-effectiveness analysis will be presented as incremental cost effectiveness ratios for our new model, demonstrating the relative cost per quality adjusted life year (QALY) through the new model. Health utilisation data captured through the impact assessment will be combined with estimated costs of health services accessed pre and post enrolment for each enrolled child. Average cost per encounter (including ED presentation, hospital admission, outpatient and GP encounters) will be applied to estimate the costs of treatment for this patient cohort and examine the differences in costs between the implemented model and traditional care.

**Data analysis.** Relevant data on the number of encounters and average costs per encounter will be obtained from the Management Support and Analysis Units in the participating LHDs. To estimate broader personal costs incurred by families, we will include survey questionnaires for families to estimate the costs associated with attending health services for their child including accommodation, parking, meals, cost of employment foregone, cost of distance travelled (per kilometre average costs for an average vehicle), and lost education opportunities (number of school days missed by the child). They will also be asked to report any out-of-pocket costs they have paid to access care or medications for their child. The expected cost impact analysis will be estimated based on the down-stream resource utilisation and funding impacts of the new model, from a health service provider and funder perspective (participating LHDs and NSW health system).

Cost-effectiveness will be measured by evaluating differences in the incremental cost-effectiveness ratio of the Asthma Care from Home model compared to the situation prior (pre-implementation) to intervention using historical data. Cost effectiveness results will be expressed as a series of incremental cost-effectiveness ratios depicting the cost per incremental change in ED presentations, hospitalisation avoided, and differences in QALYs between the two groups.

## Discussion

Asthma remains the most common chronic respiratory disease in children and adolescents in Australia and continues to have a high associated burden of care [1, 2]. Our model of care offers a solution to the current fragmented and inadequate Australian healthcare system. This model aims to standardise the care provided for children with asthma and provide easy access to in-depth educational resources and support tools for families, healthcare providers and

community services so a child's asthma can be managed effectively outside of a hospital setting and closer to home.

We have used a pre-post trial design, which will allow us to determine the changes in health services used due to our model of care. The pre-intervention period of health service utilisation for each participant will be incorporated as a temporal control. A notable strength of this design is that since the same group serves as its control, it can help control for individual differences that might confound the results in studies involving separate control groups. This can enhance the internal validity of the findings. Additionally, the gathered pre-intervention period data will incorporate five years to account for the disruption of the COVID-19 pandemic, which has been well documented to have resulted in an unseasonable decrease in asthma hospitalisations during lockdown periods [38].

In analysing our data, we will use an Interrupted Time Series Analysis (ITS) methodology. ITS analysis is one of the most effective quasi-experimental evaluation methods and has been used in many health intervention evaluations [39]. It makes full use of the longitudinal nature of the data, allowing us to control for time-varying confounders such as seasonality and co-occurring competing interventions, thereby addressing critical threats to internal validity and causal inference.

An added advantage of our methodology is integrating a theory-informed mixed-method process evaluation. This inclusion allows us to comprehensively grasp the obstacles and enablers of implementing this initiative within real-world practice, and facilitates the identification of strategies conducive to its sustained implementation [40]. Concurrently, we will undertake an economic evaluation to establish a viable rationale for healthcare organisations to allocate resources towards adopting the Asthma model of care. This will substantiate the tangible advantages of such an uptake on the Australian healthcare system.

Our trial has several limitations. First, the one-group pre-post-trial design limits our ability to assess and compare the effects of our initiative with usual care. However, the rationale for selecting this design is that the intervention involves providing a standardised asthma discharge package to eligible children, rendering the inclusion of a control waitlist impractical. After consultation with our LHD partners, a randomised controlled trial was deemed unsuitable due to the standardised nature of the intervention, thus making the proposed intervention essentially a part of standard care. We will collect and use self-reported participant data, which may introduce reporting bias. However, given the mixed-method methodology that will be employed, and the diversity of outcomes evaluated, the potential for reporting bias is minimal.

In conclusion, this study aims to significantly enhance access to asthma care, particularly targeting children in underserved communities like those living in rural and remote areas. By assessing implementation and economic factors, our research will provide comprehensive understanding of the effectiveness and viability of the model to help integrate it into standard practice. By utilising local resources, we can ensure the sustainability of improved healthcare services and help shed light on the potential for scaling up this model to benefit a wider population of children facing asthma-related challenges in underserved regions.

## Supporting information

**S1 Fig. SPIRIT schedule.**
(TIF)

**S2 Fig. Logic model.**
(TIF)

**S3 Fig. Spirit checklist.**
(TIF)

## Acknowledgments

The authors would like to thank the Sydney Children's Hospital Foundation, Asthma Australia, and Rotary Club of Sydney Cove for their continued support in our research endeavours.

This protocol has been authored on behalf of the Asthma Care from Home Collaborative Group. The authors would like to acknowledge the members of the group not listed as authors: Dr Hong Du, Dr Stuart Haggie, Mr Anthony Flynn, Ms Rose Bell, Dr Matthew O'Meara, Ms Mary Crum, Dr Corinne Langstaff, Dr Kate Molnar, Ms Tegan Zanotti, Ms Sinead Molloy, Ms Christine Dove, Ms Alanna Hoye, Ms Angela Reed, Ms Trudie Campbell, Dr Andrew Hutchinson, Dr David Meldrum, Dr James Hodges, Dr Megan Wilson, Dr Brett Shaw, Mr Tomas Ratoni, Ms Samantha Petersen, Ms Karen Greenway, Ms Tennille Carmichael, Ms Melinda Johnston, Ms Sarah Hill, Ms Ellie Saberi, Ms Karishma Behari, Ms Samantha Sweeney, Ms Tegan Moore, Ms Terry Irvine Smeets, Ms Nicole Eather, Ms Sophie Hunt, Ms Amanda Fryer, Ms Belinda Nichols, Mr Jason Connors, Ms Anneliese De Groot, Ms Tracey Tyler, Dr Kathyrn Leccese, Dr Prudence Harrison, Dr Anne Mitchell, Dr Mona Bonal, Dr Reeta Singh, Mrs Danielle Smith, Ms Meg Hale, Ms Cynthia Lloyd, Ms Julie-Ann White, Ms Linda Geale, Ms Julie Hankinson, Ms Suzanne Eddie, Ms Katherine Fisher, Ms Gretchen Buck, Ms Ainslie Tozer, Ms Felicia King, Ms Annalise Vartiainen, Ms Kelly Archer, Ms Dominique Howard, Ms, Lisa Williams, Dr Dominic Fitzgerald, Dr Julia Sgarlata, Dr Melanie Berry, Ms Rachel Edwards, Ms Kasey Streat, Ms Patricia Mannix, Ms Kayleen Cole, Ms Grace Reedy, Ms Keryn Hayes, Mr Samuel De Keizer, and Ms Prue McNamara.

## Author Contributions

**Conceptualization:** Ryan Mackle, Carmen Crespo Gonzalez, Mei Chan, Blake Angell, Louisa Owens, Jeffery Fletcher, Timothy McCrossin, Susie Piper, Aunty Kerrie Doyle, Sue Woolfenden, Bronwyn Gould, Raghu Lingam, Adam Jaffe, Melinda Gray, Nusrat Homaira.

**Funding acquisition:** Melinda Gray, Nusrat Homaira.

**Methodology:** Ryan Mackle, Carmen Crespo Gonzalez, Mei Chan, Michael Hodgins, Nan Hu, Blake Angell, Louisa Owens, Aunty Kerrie Doyle, Bronwyn Gould, Flic Ward, Raghu Lingam, Adam Jaffe, Melinda Gray, Nusrat Homaira.

**Project administration:** Ryan Mackle, Nusrat Homaira.

**Resources:** Ryan Mackle.

**Supervision:** Adam Jaffe, Melinda Gray, Nusrat Homaira.

**Writing – original draft:** Ryan Mackle, Carmen Crespo Gonzalez.

**Writing – review & editing:** Ryan Mackle, Carmen Crespo Gonzalez, Mei Chan, Michael Hodgins, Nan Hu, Blake Angell, Louisa Owens, Jeffery Fletcher, Timothy McCrossin, Susie Piper, Aunty Kerrie Doyle, Sue Woolfenden, Bronwyn Gould, Flic Ward, Raghu Lingam, Adam Jaffe, Melinda Gray, Nusrat Homaira.

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
