## [Decision Letter · Decision Letter 0]

20 Mar 2024

PONE-D-23-36574Asthma Care from Home: Study protocol for an effectiveness-implementation evaluation of a virtually enabled asthma care initiative in children in rural NSWPLOS ONE

Dear Dr. Mackle,

Thank you for submitting your manuscript to PLOS ONE. After careful consideration, we feel that it has merit but does not fully meet PLOS ONE’s publication criteria as it currently stands. Therefore, we invite you to submit a revised version of the manuscript that addresses the points raised during the review process.

**ACADEMIC EDITOR: **The submission was reviewed by the editorial board and subjected to an external review. I believe that the authors have prepared a protocol for an important research study in the field of asthma. However, they are required to address the comments sent to us by the reviewers so that weaker sections of the manuscript can be improved.

We look forward to receiving your revised manuscript.

Kind regards,

Bharat Bhushan Sharma, M.D.

Academic Editor

PLOS ONE

Journal Requirements:

2. We note that you have selected “Clinical Trial” as your article type. PLOS ONE requires that all clinical trials are registered in an appropriate registry (the WHO list of approved registries is at      https://www.who.int/clinical-trials-registry-platform/network/primary-registries"" https://www.who.int/clinical-trials-registry-platform/network/primary-registries and more information on trial registration is at http://www.icmje.org/about-icmje/faqs/clinical-trials-registration/).

Please state the name of the registry and the registration number (e.g. ISRCTN or ClinicalTrials.gov) in the submission data and on the title page of your manuscript.

a) Please provide the complete date range for participant recruitment and follow-up in the methods section of your manuscript.

b) If you have not yet registered your trial in an appropriate registry, we now require you to do so and will need confirmation of the trial registry number before we can pass your paper to the next stage of review. Please include in the Methods section of your paper your reasons for not registering this study before enrolment of participants started. Please confirm that all related trials are registered by stating: “The authors confirm that all ongoing and related trials for this drug/intervention are registered”.

Please see http://journals.plos.org/plosone/s/submission-guidelines#loc-clinical-trials for our policies on clinical trials

"This project was funded by a grant from the NSW Health Translational Research Grants Scheme. The views expressed are those of the authors and not necessarily those of the partner organisation. RM is supported by PhD scholarships from the Australian government Research Training Program (RTP), Asthma Australia and the Zoe Kennedy Foundation. "

6. We note that Figure 1 in your submission contain map image which may be copyrighted. All PLOS content is published under the Creative Commons Attribution License (CC BY 4.0), which means that the manuscript, images, and Supporting Information files will be freely available online, and any third party is permitted to access, download, copy, distribute, and use these materials in any way, even commercially, with proper attribution. For these reasons, we cannot publish previously copyrighted maps or satellite images created using proprietary data, such as Google software (Google Maps, Street View, and Earth). For more information, see our copyright guidelines: http://journals.plos.org/plosone/s/licenses-and-copyright.

Additional Editor Comments:

The submission was reviewed by the editorial board and subjected to an external review. We have received comments and suggestions by the reviewers Authors are required to address all the comments sent to us by the reviewers.

Reviewers' comments:

Reviewer's Responses to Questions

**Comments to the Author**

1. Does the manuscript provide a valid rationale for the proposed study, with clearly identified and justified research questions?

Reviewer #1: Yes

Reviewer #2: Yes

Reviewer #3: Yes

Reviewer #4: Yes

Reviewer #5: Yes

2. Is the protocol technically sound and planned in a manner that will lead to a meaningful outcome and allow testing the stated hypotheses?

Reviewer #1: Partly

Reviewer #2: Yes

Reviewer #3: Yes

Reviewer #4: Yes

Reviewer #5: Yes

3. Is the methodology feasible and described in sufficient detail to allow the work to be replicable?

Reviewer #1: Yes

Reviewer #2: Yes

Reviewer #3: Yes

Reviewer #4: Yes

Reviewer #5: Yes

4. Have the authors described where all data underlying the findings will be made available when the study is complete?

Reviewer #1: No

Reviewer #2: Yes

Reviewer #3: Yes

Reviewer #4: Yes

Reviewer #5: Yes

5. Is the manuscript presented in an intelligible fashion and written in standard English?

Reviewer #1: Yes

Reviewer #2: Yes

Reviewer #3: Yes

Reviewer #4: Yes

Reviewer #5: Yes

6. Review Comments to the Author

You may also provide optional suggestions and comments to authors that they might find helpful in planning their study.

Reviewer #1: Study is New in its concept for the individual family and the community. But without results of data analysis how can reviewer knows the effect/impact of such community based study. Control group may be considered to observe the effect whether increasing decreasing or remaining same from the same community without home based management.

Please mention the limitation of the study

Mention the exclusion criteria

how the author has avoided the selection bias

Reviewer #2: The authors of the manuscript present here the design of a quasi-experimental pre- and post-interventional trial for an asthma care from home project. This ambitious study specifically aims to improve access to asthma care for affected children and their families in rural and remote areas in Australia.

The manuscript is written with excellent expression and spelling, making it easily understandable. The authors based their design on substantial experience in former studies. Each part of the design is explained in detail and decisions made for or against certain proceedings are discussed and made transparent. One minor point from my side is to ask the authors to shortly explain why encouragment for uptake of annual influenza vaccination is implementet in the parent letter. Despite of the that, I have no further remarks.

Reviewer #3: Comments to the Authors

I. The authors provide a detailed outline of a study designed to establish a state wide program of at home asthma management for families living in rural areas. The current manuscript has two obvious problems:

i) it is too long and, in many places, repetitive.

ii) In many places the language is verbose. A particular example comes in the second page of the discussion (page 2). To this reviewer, the sentence starting on line 14 with “Moreover it facilitates” includes a remarkable number of excessive words!

II. The map does not explain the two colors if it is just to distinguish different regions it needs to say so. To a non-Australian the map would be more useful if the towns were indicated, i.e., Sydney, Newcastle, Waga-Waga, Townsville, and Brisbane.

III. There is a mention of environmental assessment but no references to the major studies on the relevance of environment, e.g., dust mites: including studies by Ann Woolcock, Jenny Peat, Dr. Marks, and Euan Tovey. All of these worked in N.S.W.

IV. Why are truly poor patients excluded? Is there a danger that those individuals will recognize that they are being excluded? How many children will be living in a shelter? What is the justification for restricting age to 5-12 years?

Minor comments

V. Do all subjects have to have cell phones?

VI. The abbreviation SMS appears in the abstract without introduction.

Reviewer #4: The study protocol aims to evaluate a digitally enabled asthma care initiative for children in rural NSW. The initiative builds on prior successful models, focusing on standardizing asthma discharge, coordinating care between acute and primary services, and providing home-based follow-up. The paper appears to be well-written. However, there are a few areas where slight improvements could be made for clarity or conciseness:

1. Rephrase this sentence "The initiative builds on prior successful models, focusing on standardizing asthma discharge, coordinating care between acute and primary services, and providing home-based follow-up,"

2. The phrase "particularly in areas with limited access to care" could be expanded for clarity.

3. The hypothesis is not correctly aligned with the study objectives and the title.

4. Sampling technique is not mentioned and how will you do the sampling.

5. Can it will possible to do the panel data modelling to include the effect of the study area?

Reviewer #5: The objective of this paper is to describe a planned intervention trial for children with asthma to improve access to asthma care for children in remote and rural areas. I commend the authors for this important work, as improving assess to care is a problem worldwide. I have a few comments on the methodology of the planned intervention:

1. Virtual home visits: only children with uncontrolled asthma, defined by ACT <19 will be offered the virtual home visit intervention. When will the ACT be administered to determine controlled/uncontrolled status? By definition, all kids presenting to the ED have current uncontrolled asthma and would likely have an ACT score at baseline <19. While this will likely lead to most children receiving the virtual home visit intervention, why not include all children?

2. Please define sever asthma in the methods.

3. Only those defined as having severe asthma will receive a telehealth consult to develop a shared asthma care plan. Why not include all children?

4. Since the children receive the virtual home visit and the telehealth based on their clinical asthma status, the study will be include multiple case groups (ie controlled non-severe, controlled severe, uncontrolled non-severe, uncontrolled severe). Since not all children are receiving the same intervention package, how will the efficacy of the different packages be determined?

5. The children will be their own control, and their previous 5 years of utilization will be used to determine if the intervention leads to decreased utilization for asthma. Since the inclusionary age starts at 5 and diagnosing asthma in children under 5 is problematic, how will it be determined if utilization below age 5 is due to true asthma and not transient wheeze?

7. PLOS authors have the option to publish the peer review history of their article (what does this mean?). If published, this will include your full peer review and any attached files.

Reviewer #1: **Yes: **Dr Lina Bandyopadhyay.

Reviewer #2: No

Reviewer #3: **Yes: **Thomas A. Platts-Mills, FRS

Reviewer #4: No

Reviewer #5: No

---

## [Author Response · Author response to Decision Letter 0]

30 Apr 2024

Dear Editor,

We are grateful to the reviewers for their review of our original protocol paper titled “Asthma Care from Home: Study protocol for an effectiveness-implementation evaluation of a virtually enabled asthma care initiative in children in rural NSW”. We have addressed the comments raised by reviewers and below are our point-by-point responses.

ACADEMIC EDITOR RESPONSE

The revised manuscript has been amended to ensure it fulfils PLOS ONE’s style requirements.

2. We note that you have selected “Clinical Trial” as your article type. PLOS ONE requires that all clinical trials are registered in an appropriate registry (the WHO list of approved registries is at https://www.who.int/clinical-trials-registry-platform/network/primary-registries"" https://www.who.int/clinical-trials-registry-platform/network/primary-registries and more information on trial registration is at http://www.icmje.org/about-icmje/faqs/clinical-trials-registration/).

Please state the name of the registry and the registration number (e.g. ISRCTN or ClinicalTrials.gov) in the submission data and on the title page of your manuscript.

a) Please provide the complete date range for participant recruitment and follow-up in the methods section of your manuscript. 

Authors response:

Thank you for your comment. The methods section has been updated according to the Editor’s suggestion.

Change(s) in manuscript:

 Methods section, Line 172-175:

“Between March 2023 and June 2024, children aged between 5-12 years old presenting to ED on one or more occasions or admitted to the general paediatric ward on one or more occasion in the last 12 months with asthma at selected study hospital sites within the three LHDs will be eligible to participate in the study.”

b) If you have not yet registered your trial in an appropriate registry, we now require you to do so and will need confirmation of the trial registry number before we can pass your paper to the next stage of review. Please include in the Methods section of your paper your reasons for not registering this study before enrolment of participants started. Please confirm that all related trials are registered by stating: “The authors confirm that all ongoing and related trials for this drug/intervention are registered”.

Please see http://journals.plos.org/plosone/s/submission-guidelines#loc-clinical-trials for our policies on clinical trials 

Authors response:

Previous correspondence detailed that clinical trial registration was not required for publication of our study protocol paper. Email thread containing this documentation has been submitted as supporting evidence (Supporting Evidence - PLOS ONE PONE-D-23-36574 - EMID6d5e96dff22f4957). 

Authors response:

We apologise for the confusion around Funding and Financial Disclosures. This study was funded by a grant from the NSW Health Translational Research Grants Scheme. These are not provided with a designated “grant number”. Our funding disclosure statement has been included in both Funding and Financial Disclosures section. RM is currently undertaking a PhD that is supported by scholarships from the Australian Government Research Training Program (RTP), Asthma Australia and the Zoe Kennedy Foundation. The PhD scholarship funding has therefore only been included within the financial disclosures section of the manuscript. 

"This project was funded by a grant from the NSW Health Translational Research Grants Scheme. The views expressed are those of the authors and not necessarily those of the partner organisation. RM is supported by PhD scholarships from the Australian government Research Training Program (RTP), Asthma Australia and the Zoe Kennedy Foundation. "

Authors response:

We have amended the financial disclosure statement with this submission. Please see cover letter.

Authors response:

Thank you for your comment. The full ethics statement has now been added to the ‘Methods’ section of the revised manuscript according to the Editor's suggestions.

Change(s) in manuscript:

Study design and setting section, Line 166-170: 

“Ethical approval was granted by the Sydney Children’s Hospital Network Human Research Ethics Committee in November 2022 (2022/ETH01946). Site specific Research Governance was granted across all participating Local Health District Research Governance by March 2023 (2022/STE03422, 2022/STE03423, 2022/STE03424 and 2022/STE03425).” 

Study population section, Line 184-186:

“Informed written consent was sought from parents/carers of eligible children electronically via REDcap software.”

6. We note that Figure 1 in your submission contain map image which may be copyrighted. All PLOS content is published under the Creative Commons Attribution License (CC BY 4.0), which means that the manuscript, images, and Supporting Information files will be freely available online, and any third party is permitted to access, download, copy, distribute, and use these materials in any way, even commercially, with proper attribution. For these reasons, we cannot publish previously copyrighted maps or satellite images created using proprietary data, such as Google software (Google Maps, Street View, and Earth). For more information, see our copyright guidelines: http://journals.plos.org/plosone/s/licenses-and-copyright.

Authors response:

The map included in our original manuscript was adapted from a map included in a previous journal manuscript submitted by our team (Chan et al, Journal of Asthma and Allergy 2021 (DOI: 10.2147/jaa.s311721)). As it does not fulfil PLOS One guidelines, we have removed it from our revised manuscript.

We have made the requested amendments to Supporting Information 

Authors response:

Thank you for your comment. The reference lists have been reviewed and updated to improve accuracy.

Change(s) in manuscript:

Reference list section:

1 Global Asthma Network. The Global Asthma Report 2022 Auckland. New Zealand: Global Asthma Network; 2022. [cited 2022 28 August]. Available from: www.globalasthmareport.org/resources/global_asthma_report_2022.pdf. 

Changed to 

1. Global Asthma Network. The Global Asthma Report 2022. Auckland, New Zealand 2022 [cited 2024 26th April]. Available from: http://globalasthmareport.org

2 Global Asthma Network. The Global Asthma Report 2018 Auckland. New Zealand: Global Asthma Network; 2018. [cited 2023 28 August]. Available from: www.globalasthmareport.org/resources/global_asthma_report_2018.pdf 

Changed to

2. Global Asthma Network. The Global Asthma Report 2018. Auckland, New Zealand 2018 [cited 2024 26th April]. Available from: http://globalasthmareport.org/2018/index.html

6. HealthStats. NSW Population by remoteness, age and year 2020

Changed to

Centre for Epidemiology and Evidence. Asthma prevalence in children by Asthma Type and Remoteness category. Sydney: HealthStats NSW; 2020 [cited 2024 26th April]. Available from: https://www.healthstats.nsw.gov.au/r/112635

9. Australian Bureau of Statistics. QuickStats,. Canberra,, 2021.

Changed to:

7. Australian Bureau of Statistics. Quickstats, Australia. Canberra: Australian Bureau of Statistics; 2021.

17. NSW Health. Schools and child services action plan for asthma flare-up.: NSW Government; 2023 [Available from: https://www.schn.health.nsw.gov.au/files/factsheets/asthma_-_schools_and_child_services_action_plan_for_asthma_flare_up-en.pdf]. 

Changed to 

19. NSW Health. Schools and Child Services Action Plan for Asthma Flare-up. [cited 2024 26th April]. Available from: https://www.schn.health.nsw.gov.au/sites/default/files/2024-01/21138465_schn_asthma_flare_up_document_v3_1.pdf

New reference in response to reviewers comments:

18. Department of Health and Aged Care. Influenza (Flu), Australian Immunisation Handbook. [cited 2024 26th April]. Available from: https://immunisationhandbook.health.gov.au/

20. Casale TB, Pedersen S, Rodriguez Del Rio P, Liu AH, Demoly P, Price D. The Role of Aeroallergen Sensitization Testing in Asthma Management. J Allergy Clin Immunol Pract 2020;8(8):2526-32

35.Weiner BJ, Lewis CC, Stanick C, Powell BJ, Dorsey CN, Clary AS, Boynton MH, & Halko H. Psychometric assessment of three newly developed implementation outcome measures. Implementation Science, 2017;12(108), 1-12.

 

REVIEWER RESPONSE

Reviewer #1: 

Study is New in its concept for the individual family and the community. But without results of data analysis how can reviewer knows the effect/impact of such community-based study. Control group may be considered to observe the effect whether increasing decreasing or remaining same from the same community without home-based management.

Please mention the limitation of the study 

Mention the exclusion criteria 

how the author has avoided the selection bias. 

Authors response:

We appreciate the reviewer’s comment. As this is a study protocol paper, hence does not describe any result, this paper outlines how the study will be conducted. We will have the results once the study is completed. We do accept that the one-group pre-post-trial design limits our ability to assess and compare the effects of our initiative with usual care, however in consultation with our LHD partners, a randomised controlled trial was deemed unsuitable due to the standardised nature of the intervention. Instead, we chose a pre/post quasi-experimental study design whereby each participant acts as their own control which minimises selection bias. As we are including all participants who meet eligibility criteria, convenience and random sampling are not indicated. Exclusion criteria is available in the Methods section [Line 196-203 original manuscript] and study limitations available in the Discussion [Line 517-526 original manuscript]. 

Reviewer #2: 

The authors of the manuscript present here the design of a quasi-experimental pre- and post-interventional trial for an asthma care from home project. This ambitious study specifically aims to improve access to asthma care for affected children and their families in rural and remote areas in Australia.

The manuscript is written with excellent expression and spelling, making it easily understandable. The authors based their design on substantial experience in former studies. Each part of the design is explained in detail and decisions made for or against certain proceedings are discussed and made transparent. One minor point from my side is to ask the authors to shortly explain why encouragement for uptake of annual influenza vaccination is implemented in the parent letter. Despite of the that, I have no further remarks.

Authors response:

We thank Reviewer 2 for their valuable feedback. Annual influenza vaccination is recommended in international guidelines due to the increased risk of morbidity and mortality associated with influenza infection in asthmatics. The Australian Immunisation Handbook also recommends seasonal influenza vaccination for children with asthma. The manuscript has been amended to make our rationale clearer.

Change(s) in manuscript:

Study intervention section, Line 209-211:

“…a parent letter encouraging regular GP follow-up and uptake of annual influenza vaccination (as recommended in GINA guidelines [17] and Australian Immunisation Handbook [18].”

Reviewer #3: 

I. The authors provide a detailed outline of a study designed to establish a state wide program of at home asthma management for families living in rural areas. The current manuscript has two obvious problems:

i) it is too long and, in many places, repetitive.

ii) In many places the language is verbose. A particular example comes in the second page of the discussion (page 2). To this reviewer, the sentence starting on line 14 with “Moreover it facilitates” includes a remarkable number of excessive words! [Line 514-517]

Authors response:

We appreciate Reviewer 3’s feedback. We have updated the manuscript to make it more concise.

Change(s) in manuscript:

Study interventions section, Line 205-211:

“Participants discharged from the hospital will receive the “Asthma Resource Discharge package” which contains several pre-developed resources including an individualised asthma action plan (AAP) in colour [16], an asthma education booklet entitled “Asthma and Your Child – a resource pack for parents and carers” to improve parent’s/carer’s understanding of asthma that is available in multiple languages, a parent letter encouraging regular GP follow-up and uptake of annual influenza vaccination (as recommended in Global Initiative for Asthma (GINA) guidelines [17] and the Australian Immunization Handbook [18])”

Study interventions section, Line 222-231:

“Parents/carers of the participating child will firstly receive a text message (SMS) reminder within 7 days of hospital discharge to remind them of regular GP follow-up and provide links to online resources and links to register for monthly educational webinars with the AAIC. In addition, local investigators will coordinate follow-up appointments with GPs and/or paediatricians through telephone or email to facilitate tele-consultation where possible and provide discharge instructions. Similarly, the child’s school or childcare service will be notified by email of the child’s recent asthma hospitalisation, provided an 

---

## [Decision Letter · Decision Letter 1]

17 May 2024

Asthma Care from Home: Study protocol for an effectiveness-implementation evaluation of a virtually enabled asthma care initiative in children in rural NSW

PONE-D-23-36574R1

Dear Dr. Mackle,

We’re pleased to inform you that your manuscript has been judged scientifically suitable for publication and will be formally accepted for publication once it meets all outstanding technical requirements.

Kind regards,

Bharat Bhushan Sharma, M.D.

Academic Editor

PLOS ONE

Additional Editor Comments (optional):

The changes done in the manuscript are satisfactory.

Reviewers' comments:

Reviewer's Responses to Questions

**Comments to the Author**

1. Does the manuscript provide a valid rationale for the proposed study, with clearly identified and justified research questions?

Reviewer #1: Yes

Reviewer #3: Yes

Reviewer #5: Yes

2. Is the protocol technically sound and planned in a manner that will lead to a meaningful outcome and allow testing the stated hypotheses?

Reviewer #1: Yes

Reviewer #3: Yes

Reviewer #5: Yes

3. Is the methodology feasible and described in sufficient detail to allow the work to be replicable?

Reviewer #1: Yes

Reviewer #3: Yes

Reviewer #5: Yes

4. Have the authors described where all data underlying the findings will be made available when the study is complete?

Reviewer #1: Yes

Reviewer #3: Yes

Reviewer #5: Yes

5. Is the manuscript presented in an intelligible fashion and written in standard English?

Reviewer #1: Yes

Reviewer #3: Yes

Reviewer #5: Yes

6. Review Comments to the Author

You may also provide optional suggestions and comments to authors that they might find helpful in planning their study.

Reviewer #1: Satisfactory information being obtained as per reviewers queries .Protocol may be accepted for future activities.

Reviewer #3: The authors have responded well to our comments. In particular, the modifications to the text represent a considerable improvement. Overall, the manuscript is much improved. I have two minor comments.

i) I was disappointed that the map was removed. I understand that the editors made extensive suggestions about it. However, for a reader in the UK or the USA, it would be difficult to appreciate what was being proposed without the map. I have commented to the editor. In particular, it is difficult to appreciate the distances involved without a map.

ii) I am afraid the authors assumed I was an Australian. I suggested the Australian authors who had worked on the relevance of dust mite exposure in asthma. I am aware of the Australian work and believe there is some important work from Dr. Tovey on the effect of very high mite levels in N.S.W. (See Tovey ER, Almqvist C, Li Q, Crisafulli D, Marks GB. Nonlinear relationship of mite allergen exposure to mite sensitization and asthma in a birth cohort. J All Clin Immunol 2008; 121 :114-118)

Reviewer #5: The authors have addressed all of my concerns. I have no further comments or suggestions at this time.

7. PLOS authors have the option to publish the peer review history of their article (what does this mean?). If published, this will include your full peer review and any attached files.

Reviewer #1: **Yes: **Dr Lina Bandyopadhyay

Reviewer #3: **Yes: **Thomas A. E. Platts-Mills

Reviewer #5: No

---

## [Editor Report · Acceptance letter]

22 May 2024

PONE-D-23-36574R1 

PLOS ONE

Dear Dr. Mackle, 

I'm pleased to inform you that your manuscript has been deemed suitable for publication in PLOS ONE. Congratulations! Your manuscript is now being handed over to our production team.

Kind regards, 

on behalf of

Professor Bharat Bhushan Sharma 

Academic Editor

PLOS ONE